# A High-Precision Hand–Eye Coordination Localization Method under Convex Relaxation Optimization

**DOI:** 10.3390/s24123830

**Published:** 2024-06-13

**Authors:** Jin Hua, Yuhang Su, Daxin Xin, Weidong Guo

**Affiliations:** School of Electronic Information Engineering, Xi’an Technological University, Xi’an 710021, China; 15319223809@163.com (Y.S.); xdx661006@163.com (D.X.); 13186269094@163.com (W.G.)

**Keywords:** switching operation, visual servo, convex relaxation, hand–eye coordination

## Abstract

Traditional switching operations require on-site work, and the high voltage generated by arc discharges can pose a risk of injury to the operator. Therefore, a combination of visual servo and robot control is used to localize the switching operation and construct hand–eye calibration equations. The solution to the hand–eye calibration equations is coupled with the rotation matrix and translation vectors, and it depends on the initial value determination. This article presents a convex relaxation global optimization hand–eye calibration algorithm based on dual quaternions. Firstly, the problem model is simplified using the mathematical tools of dual quaternions, and then the linear matrix inequality convex optimization method is used to obtain a rotation matrix with higher accuracy. Afterwards, the calibration equations of the translation vectors are rewritten, and a new objective function is established to solve the coupling influence between them, maintaining positioning precision at approximately 2.9 mm. Considering the impact of noise on the calibration process, Gaussian noise is added to the solutions of the rotation matrix and translation vector to make the data more closely resemble the real scene in order to evaluate the performance of different hand–eye calibration algorithms. Eventually, an experiment comparing different hand–eye calibration methods proves that the proposed algorithm is better than other hand–eye calibration algorithms in terms of calibration accuracy, robustness to noise, and stability, satisfying the accuracy requirements of switching operations.

## 1. Introduction

With social and economic development and the continuous expansion of the scale of enterprise, the electrical maintenance workload is increasing daily. Traditional switching operations [1,2,3] have high risk factors and low efficiency, creating an urgent need for new technology to change the mode of operation, reduce the burden on staff, and ensure the safe and stable operation of electrical equipment in distribution rooms. In recent years, with the rapid development of artificial intelligence and robotics technology, intelligent robots [4,5,6] for power distribution rooms have emerged. Through the integration of a six-degree-of-freedom mechanical arm, 3D depth cameras, and other means, most manual operations and maintenance tasks can be performed by robots. Operators can remotely monitor the status of a robot and control it to achieve remote operation, maximizing unmanned distribution room management, reducing labor costs, minimizing safety risks, improving the efficiency of switching operations, and realizing intelligent construction.

The specific tasks of the switching operation referred to in this article include opening power meter knobs, opening and closing secondary switches, and inserting and removing keys. Precise localization of these actions can be accomplished by combining a depth camera with a robot. The camera can be placed outside the robot’s body (eye-to-hand) or mounted on the robot as part of the robot’s end-effector (eye-in-hand) [7,8,9]. Regardless of where the camera is placed, the position of the target object is initially obtained by the camera’s coordinate system and needs to be transformed into the robot’s base coordinate system. This requires an accurate estimation of the relationship between the two coordinate systems (the transformation matrix), also known as hand–eye calibration. The accuracy of the hand–eye calibration results directly affects whether the robot can accurately perform tasks such as the localization and grasping of target objects. Due to the nonlinearity of the solution to this problem, obtaining the optimal solution through optimization techniques is a key focus for researchers. Only through scientific and effective hand–eye calibration algorithms can we ensure accurate matching between the robot and the vision system, thereby improving the recognition and localization of target objects.

Tsai et al. [10] and Shiu et al. [11] were the first to pose the hand–eye calibration problem and simplified it into solving the AX = XB equation, where the hand–eye matrix X to be solved contains a 3 × 3 dimensional rotational component matrix R and a 3 × 1 dimensional translation matrix T. This is a mutually coupled nonlinear solution, and the normally solved X has multiple solutions that cannot be uniquely determined. Due to this nonlinear problem, researchers have proposed different methods to solve this equation, roughly categorized into two types: The classical hand–eye calibration method proposed by Tsai decomposes the problem into two parts, the rotation matrix and the translation vector, solving for the rotation matrix first and then for the translation vector. The two-step solution method has a simple solution process and is insensitive to the noise of the measured data in the translation part, but the solution accuracy of the translation vectors is susceptible to rotation matrix error transfer. Researchers such as Frank and Navy [12] proposed a hand–eye calibration algorithm, and Daniilidis K [13] proposed the Dual-Quaternion (DQ) method. These algorithms parameterize the calibration equations and solve them with the help of mathematical tools such as rotation vectors, Lie group–Lie algebra, unit quaternion, and dual quaternion. However, these methods are affected by direction-to-position error propagation due to external noise interference in the actual measurement, which affects the stability of the solution accuracy. Cui et al. [14] came up with a new two-step hand–eye calibration method, simplifying the calibration process by making the robot perform specified motions to obtain the hand–eye matrix. Andreff et al. [15] proposed a method of solving the basic matrix of hand–eye calibration using the theory of the matrix direct product, namely, using the direct product of the matrix to solve the basic matrix of hand–eye calibration. In this method, the basic matrix is written in a linearized form, and then the linearization method is used to solve it. This linearized solution method is simple and easy to use and effectively avoids the possible solution instability of the nonlinearized method. Junchen Wang et al. [16] proposed a nonlinear optimization algorithm based on maximum likelihood estimation in the Eulerian rigid variation matrix space, reducing the distance between the true value and the corresponding measured value to improve stability and accuracy. Jinqiao Wang et al. [17] used a genetic algorithm to obtain the initial value of the hand–eye transformation matrix using conventional methods and then optimized the mathematical model based on the initial value, using the genetic algorithm to achieve higher measurement accuracy. Zhaorui Zhang et al. [18] proposed a constraint matrix construction method that integrates two kinds of motion information, analyzed the constraint matrix rank, and constructed the constraint matrix to ensure its complete ranking according to different situations, calculating the rotation matrix and translation vector. In general, there is relatively little comprehensive research on the algorithms for solving the calibration equations and the factors affecting the errors, both domestically and internationally. In fact, the screening of calibration data and the algorithm used for solving the calibration equations directly affect the accuracy of solving the hand–eye transformation matrix, so it is necessary to comprehensively study these factors.

Aiming at the above problems of nonlinearity and coupling in the hand–eye calibration solution, this article proposes a method for addressing the key problems in robot hand–eye calibration. This method first establishes the hand–eye calibration problem using pairwise quaternions through the concept of functions. Secondly, a convex relaxation global optimization algorithm based on linear matrix inequality is introduced for the rotation and translation solutions in the hand–eye calibration problem. This approach overcomes the limitations of traditional nonlinear optimization algorithms, which rely on the initial value and easily fall into local optima. In conclusion, through experimental verification, it is proven that the algorithm proposed in this paper has better solution accuracy and stability compared to traditional hand–eye calibration algorithms, enabling the accurate positioning of instrument knobs, secondary switches, and locking holes on the electric control cabinets in power distribution rooms.

## 2. Coordinated Hand–Eye Reverse Gate Operation

### 2.1. Eye-in-Hand System Structure Design

The traditional manual implementation of the corresponding switching operation requires coordination and cooperation. The long operating time involved in this process, accompanied by arc discharge, can expose the operator to the risk of accidental injury, making it difficult to complete the corresponding task in the power system in a stable manner. Therefore, this study uses a robotic arm instead of a human hand to perform the corresponding inverted gate task and a depth camera to simulate the human eye to observe the specific location of the object to realize the automated operation process. As shown in Figure 1, the robot has three overall components:(1)A six-degree-of-freedom robotic arm allows highly accurate control and can ensure the accuracy of the reverse gate operation, safeguarding against operation failure and equipment damage. In some narrow and difficult-to-access spaces, the six-degree-of-freedom robotic arm can flexibly maneuver the inverting lever for the purpose of opening or closing the gate blade without the need for the operator to physically enter the narrow or dangerous area.(2)A depth camera, commonly using infrared or laser sensors, captures the three-dimensional information of a scene, allowing for the more accurate determination of the distance and shape of objects. Real-time video streaming can be used to monitor the progress of the reversing operation and the status of the equipment, helping the operator to pinpoint the location and position of the reversing equipment for the proper placement of switch levers, handles, or other controls(3)A uniquely designed jaw provides high-precision control, ensuring the accuracy of the reverse gate operation and reducing the risk of misuse. It is intended for use in rotating reversing gate equipment to ensure that meter buttons are opened or closed correctly. It can control the rotary movement via a robotic arm or other devices to precisely control the operation. Generally speaking, the eye-in-hand system adopts the eye-in-hand mounting method, where the camera is fixed on the end-effector gripper. This method provides relative flexibility and allows the camera to be moved with the robot for image acquisition, and the distance of the camera can be adjusted when facing target objects of different sizes to reduce measurement errors and improve accuracy.

As illustrated in Figure 1, the positioning of the object needs to be performed through the following steps. First, the depth camera captures the position information of the object, identifying its features and determining the pixel coordinates in the image to complete the specific task of positioning. Then, the two-dimensional image coordinates are converted to three-dimensional coordinates. Using the transformation matrix between the coordinate system of the end effector’s clamping claws and the camera coordinate system, the robot arm performs the specific task based on the object’s position. Next, the kinematics of the robot arm are solved, calculating the object’s position. Finally, the position of the object is communicated to the robotic arm, enabling it to move to the appropriate position to complete the specific switching operation.

### 2.2. Description of the Hand–Eye Calibration Problem

When a robot performs a task under the guidance of vision, it is necessary to know the relative positions of the robot end effector and the target to facilitate the subsequent execution of the corresponding task. A camera is fixed to the robot end effector as a vision sensor. The spatial position of the target workpiece relative to the camera is known, and the position of the camera relative to the end of the robotic arm needs to be determined in order to accurately identify and localize the target workpiece within the robot’s coordinate system. The robot vision system in Figure 1 consists of a robot and a camera mounted on an end effector, recording a set of relative robot motions. There is no change in the relative positions of the robot base and the calibration plate, nor in the relative positions of the camera and the robot end effector. Based on the multiple sets of known invariants, the hand–eye calibration matrix can be solved:(1)Tg1b⋅Tcg⋅Tt1c=Tg2b⋅Tcg⋅Tt2cTg2b−1⋅Tg1b⋅Tcg=Tcg⋅Tt2c⋅Tt1c−1LetTg2b−1⋅Tg1b=A,Tt2c⋅Tt1c−1=B,Tcg=X,then:AX=XB
where Tgb denotes the chi-squared matrix of the robot end coordinate system with respect to the base coordinate system; Ttc denotes the sub-matrix of the calibration plate coordinate system with respect to the camera coordinate system; Tcg denotes the sub-matrix of the camera coordinate system with respect to the robot end coordinate system; *A* denotes the relative attitude between two movements of the camera; *B* denotes the relative attitude between two movements of the robot end effector; and *X* denotes the relative attitude of the camera with respect to the end effector. This equation can be rewritten in another form:(2)$$\left\{\begin{array}{l} A=A_{1}A_{2}^{-1}\\ B=B_{2}B_{1}^{-1} \end{array}\right.$$
where $A_{1}$ and $A_{2}$ denote the position matrices of the camera relative to the target object with respect to the two motions of the camera; $B_{1}$ and $B_{2}$ denote the position matrices of the base relative to the end with regard to the two motions of the robot end-effector. For the camera, matrix *A* can be obtained from the external parameter calibration of the camera. Matrix *B* can be obtained from the forward kinematics equations of the robot or read out from the robot controller output. The hand–eye calibration equations can be decoupled into rotational and translational parts if matrices *A*, *X*, and *B* are assumed to consist of the corresponding rotational and translational parts:(3)$$A=\left[\begin{array}{cc} R_{A} & t_{A}\\ 0 & 1 \end{array}\right],B=\left[\begin{array}{cc} R_{B} & t_{B}\\ 0 & 1 \end{array}\right],X=\left[\begin{array}{cc} R_{X} & t_{X}\\ 0 & 1 \end{array}\right]$$

Then, Equation (1) can be transformed into
(4)$$\left[\begin{array}{cc} R_{A} & t_{A}\\ 0 & 1 \end{array}\right]\left[\begin{array}{cc} R_{X} & t_{X}\\ 0 & 1 \end{array}\right]=\left[\begin{array}{cc} R_{X} & t_{X}\\ 0 & 1 \end{array}\right]\left[\begin{array}{cc} R_{B} & t_{B}\\ 0 & 1 \end{array}\right]$$

Further expansion of this chi-squared equation yields the rotation vector equation and the translation vector equation:(5)$$\left\{\begin{array}{l} R_{A}R_{X}=R_{X}R_{B}\\ R_{A}t_{X}+t_{A}=R_{X}t_{B}+t_{X} \end{array}\right.$$

Solving the two equations of the above equation yields the solution to equation AX=XB, the general form of the unfolding of the model of the hand–eye calibration problem. At least two position changes (and thus at least three positions of the camera calibration results) are needed to solve this problem. The above process of solving linear equations is only carried out under the assumption that all the parameters are known to satisfy the condition that there is a unique solution to the equation. However, in real life, there is no ideal hypothesis, and we need to realistically solve the positional relationship between the target object and the robot in order to accurately perform a task. Currently, robots are used in more and more industries and perform tasks for a wide variety of objects, but the hand–eye transformation matrix is solved uniquely for different objects. The presence of multiple solutions leads to non-unique convergence of the results unfavorable for end-effector operation.

For the switching operation discussed in this article, accurately locating the positions of the instrument knob, secondary switch, and lock hole on the electric control cabinet is crucial. Each task scenario varies in terms of its execution requirements. Traditional calibration algorithms may tolerate small errors in tasks with low precision requirements, such as positioning the instrument knob or secondary switch. In such cases, the robot can still complete the task. However, tasks involving the positioning of the lock hole demand high-precision calibration. Traditional optimization algorithms often rely on initial value selection, which can prematurely converge to a local optimum, leading to task execution instability. To address this, we propose the idea of global optimization, in which all possible solutions of the function within given constraints are searched for, and the optimal solution is selected for transmission to the robot.

## 3. Dual Quaternions for Solving Hand–Eye Calibration Problems

### 3.1. Dual Quaternions

Dual quaternions are an extended form of quaternions used to represent rigid body transformations and animation interpolation. They are widely used in computer graphics and robotics. A quaternion is a mathematical object consisting of one real part and three imaginary parts, usually denoted as g=a+bi+cj+dk, where *a*, *b*, *c*, and *d* are real numbers, while *i*, *j*, and *k* are imaginary units that satisfy the following relation: $i^{2}=j^{2}=k^{2}=ijk=-1$. Quaternions have the advantage of rotational representation and are used more often in rotational operations in three-dimensional space.

The dual quaternion introduces another quaternion as its counterpart on top of the quaternion. It has a wide range of applications in object position measurement and can be used to solve rotational relationships effectively. A dual quaternion can be understood to be a pair of quaternions whose elements are dual quaternions or a quaternion whose elements are dual quaternions. According to the first conceptualization, dual quaternions can be expressed in the following form:(6)$$\hat{g}=g+\varepsilon g\prime =\left(\begin{array}{c} g_{0}\\ \vec{g} \end{array}\right)+\varepsilon \left(\begin{array}{c} {g^{\prime }}_{0}\\ \vec{g^{\prime }} \end{array}\right)$$
where g,g′ denote a pure quaternion ($g_{0}\;\mathrm{and}\;{g\prime }_{0}$ are the actual values); $\vec{g},\vec{g\prime }$ denote the real part (the non-dual part) and the imaginary part (the dual part) of the dual quaternion, $\vec{g}={\left[\begin{array}{ccc} g_{1} & g_{2} & g_{3} \end{array}\right]}^{T},\vec{g^{\prime }}{\left[\begin{array}{ccc} {g_{1}}^{\prime } & {g_{2}}^{\prime } & {g_{3}}^{\prime } \end{array}\right]}^{T}$; and *ɛ* is the calibration constant ($\varepsilon^{2}=0$ but ε≠0). The corresponding conjugate of a dual quaternion is defined as follows:(7)$${\hat{g}}^{*}=g^{*}+\varepsilon {g^{\prime }}^{*}=\left(\begin{array}{c} g_{0}\\ -\vec{g} \end{array}\right)+\epsilon \left(\begin{array}{c} {g^{\prime }}_{0}\\ -\vec{g\prime } \end{array}\right)$$

For any two pairs of even numbers $\hat{x}$ and $\hat{y}$, the main operation is
(8)$$\hat{x}+\hat{y}=\left(x+y\right)+\varepsilon \left(x^{\prime }+y^{\prime }\right)\;\lambda \hat{x}=\lambda x+\lambda \varepsilon x^{\prime }\;\hat{x}\cdot \hat{y}=\left(x\cdot y\right)+\varepsilon \left(y\cdot x^{\prime }+x\cdot y^{\prime }\right)$$

According to the definition of a dual quaternion, the unit dual quaternion satisfies the following conditions:(9)$${\hat{g}}^{*}\cdot \hat{g}=\left(\begin{array}{c} 1\\ 0 \end{array}\right)$$

Namely, the real part of the unit dual quaternion is the unit quaternion:(10)$$g^{*}\cdot g=\left(\begin{array}{c} 1\\ 0 \end{array}\right)$$

The dyadic part satisfies the orthogonality condition with respect to the real part:(11)$$g_{0}{g_{0}}^{\prime }+<\vec{g,}\vec{g\prime }>=0$$

### 3.2. Solving the Calibration Equation Using Dual Quaternions

The calibration process requires at least two non-parallel rotational axes to obtain the *i* bit-position transformed observation equations, and substituting them in dual quaternion form into equation *AX* = *XB* yields
(12)$${\hat{x}}_{i}\cdot {\hat{g}}_{x}={\hat{g}}_{x}\cdot {\hat{y}}_{i}$$
where the dual quaternion ${\hat{x}}_{i}=x_{i}+x_{i}\prime ,{\hat{y}}_{i}=y_{i}+y_{i}\prime$ is denoted by A and B in Equation (1) in the *i*-th bit-pose transformation, respectively; the dual quaternion ${\hat{g}}_{x}=g_{x}+{g^{\prime }}_{x}$ is denoted by X in Equation (1), namely, the required hand–eye relation matrix X. Equation (12) can be expanded as follows:(13)$$\left\{\begin{array}{l} x_{i}\cdot g_{x}-g_{x}\cdot y_{i}=0\\ \left(x_{i}\prime \cdot g_{x}-g_{x}\cdot y_{i}\prime \right)+\left(x_{i}\cdot g_{x}\prime -g_{x}\prime \cdot y_{i}\right)=0 \end{array}\right.$$

According to Chen’s theory of spiral motions (namely, describing motions in terms of dual quaternions), when the scalar equations are expressed as in Equation (13), the scalar part of $\hat{x}$ and the scalar part of $\hat{y}$ are equal. Using the property wherein the rotation angles and translation distances of motions A and B are equal, six equations can be obtained by removing the redundant equations. These six equations are written as a matrix as follows:(14)$$\left[\begin{array}{cccc} x-y & {\left[x+y\right]}_{\times } & 0_{3\times 1} & 0_{3\times 3}\\ x^{\prime }-y^{\prime } & {\left[x^{\prime }+y^{\prime }\right]}_{\times } & x-y & {\left[x+y\right]}_{\times } \end{array}\right]\cdot \left[\begin{array}{c} t\\ t^{\prime } \end{array}\right]=0$$

There are eight unknowns in Equation (14), and by denoting the 6 × 8 matrix on the left-hand side of the equation as S, the following 6i×8 matrix *S* can be constructed for *i* motions:(15)$$T={\left(S_{1}^{T},S_{2}^{T},\cdots ,S_{i}^{T}\right)}^{T}$$

The singular value decomposition of the matrix *T* in Equation (15) is shown below:(16)$$SVD\left(T\right)=U\sum V^{T}$$
where U and V are the left singular matrix and right singular matrix, respectively; ∑ is the diagonal array of singular values. In the absence of noise interference, the matrix T attains a complete rank, and this rank is six. Then, $\left[\begin{array}{c} g\\ g^{\prime } \end{array}\right]$ must form a linear combination with the last two vectors $V_{7}$ and $V_{8}$ of the right singular matrix V; from this, we can obtain the real part of the solution for the corresponding rotated part of the dual quaternion and then substitute it into Equation (5) to find the translation vector t.

## 4. Convex Relaxation Global Optimization Algorithm for Solving Hand–Eye Calibration Equations

### 4.1. Convex Relaxation Global Optimization Algorithm

The convex relaxation global optimization algorithm is a method used to solve global optimization problems. A global optimization problem involves finding the globally optimal solution to a function under given constraints, aiming to minimize or maximize the objective function [19,20,21]. The convex relaxation algorithm, rooted in convex optimization theory, tackles the original non-convex global optimization problem by transforming it into an equivalent convex optimization problem. This method offers several advantages, as convex optimization problems are relatively easier to solve, with numerous efficient algorithms and tools available for this purpose. The linear matrix inequality (LMI) solution, grounded in convex optimization theory, ensures the existence and feasibility of the global optimal solution. Optimization using the LMI relaxation technique [22] is tailored for convex relaxation polynomial optimization problems, characterized by the absence of initial value estimation. Theoretically, the LMI method stands as one of the most reliable choices, maximizing the assurance of computationally searching for the global optimum. Hence, in this article, we convert the optimization problem in the hand–eye transformation matrix into a convex relaxation polynomial optimization problem to pursue the optimal solution. 

Setting a scalar multivariate polynomial over $W\left(x\right)\;is:x=\left(x_{1},x_{2},\cdots x_{m}\right)\in C^{m}$, the optimization problem for multivariate polynomials can usually be described as follows:(17)$$\min W\left(x\right)\;s.t.D_{i}\left(x\right)\geq 0,i=1,2,\cdots ,k\;Among:x={\left(x_{1},x_{2},\cdots ,x_{m}\right)}^{T}\in C^{m}\;W\left(x\right)\in C^{m}\;D_{i}\left(x\right)\in K^{ni}\left[x\right]$$
where $W\left(x\right)$ and $D_{i}\left(x\right)$ both denote multivariate polynomials associated with *x*; $K^{ni}\left[x\right]$ denote the set of $n_{i}\times n_{i}$ symmetric matrices with polynomial entries; and $D_{i}\left(x\right)\geq 0$ show that the constraints are semipositive definite. Assuming that the number of highest-order terms of $W\left(x\right)$ and $D_{i}\left(x\right)$ is a known value, the above problem becomes a convex linear matrix inequality (LMI) optimization problem.

Lasserre [23] gives a general form for solving the LMI optimization problem. First, define the multivariate polynomial $D\left(x\right)$ as
(18)$$D\left(x\right)=\sum_{\alpha \leq t}p_{\alpha }x_{1}^{\alpha }x_{2}^{\alpha }\cdots x_{m}^{\alpha }=\sum_{\alpha \leq t}p_{\alpha }\psi \left(x\right)$$
where $p_{\alpha }$ is the vector of coefficients of $D\left(x\right)$; $\psi \left(x\right)$ is a canonical basis for $d=\left[\begin{array}{c} m+n\\ m \end{array}\right]$ monomials.
(19)$$\psi \left(x\right)=\left(1,x_{1},x_{2},\cdots ,x_{m},x_{1}^{2},x_{1}x_{2},x_{2}x_{3},\cdots \right)$$

In order to construct the Lasserre hierarchy, it is necessary to determine the minimum relaxation factor $t_{\min}$:(20)$$t_{\min}=\max\left\{1,\frac{degF\left(x\right)}{2},\frac{degD_{1}\left(x\right)}{2},\cdots ,\frac{degD_{k}\left(x\right)}{2}\right\}$$
where: *deg* denotes the number of highest-order terms of the polynomial.

Then, a linear function $L_{y}$ is introduced to linearize the multivariate polynomial involved in the LMI optimization problem by replacing the monomial $x^{\alpha }$ with a new variable $y_{\alpha }$. The linearized objective function is
(21)$$L_{y}\left[F\left(x\right)\right]=\sum_{\alpha \leq t}p_{a}y_{\alpha }$$

The constraints after linearization are
(22)$$M_{n}\left(y\right)=L_{y}\left[\psi \left(x\right)\psi {\left(x\right)}^{T}\right]\;M_{n}\left[D_{i}\left(x\right),y\right]=L_{y}\left[\psi \left(x\right)\psi {\left(x\right)}^{T}⊙D_{i}\left(x\right)\right]$$
where ⊙ denotes the Kronecker matrix product.

Finally, the above global optimization problem is modeled as an LMI optimization model:(23)$$\min L_{y}\left[D\left(x\right)\right]\;s.t\;M_{t}\left(y\right)\geq 0\;i=1,\cdots ,k\;M_{t-d}\left(y\right)\geq 0\;i=1,\cdots ,k\;Among:\;y={\left(y_{1},y_{2},\cdots ,y_{d}\right)}^{T}\in C^{m}\;d_{i}=\left[\frac{degD_{i}\left(x\right)}{2}\right],\;d=\left[\begin{array}{c} m+2t\\ m \end{array}\right]$$

Assuming that there exists an optimal solution $y^{*}$, it can be verified that the final result is globally optimal according to the matrix rank equality condition.
(24)$$rank\left[M_{t-d}\left(y^{*}\right)\right]=rank\left[M_{t}\left(y^{*}\right)\right]$$

The optimal solution of the above problem, $y^{*}$, can be obtained by invoking linear optimization solvers such as CSDP, SDPA, and Sedumiand, among other toolboxes.

### 4.2. Convex Relaxation Optimization for Solving Hand–Eye Calibration Equations

Based on the quadratic number multiplication property, for unit quaternions *r* and *g*, we have
(25)$$r\cdot g=A\left(r\right)g=B\left(g\right)r$$
where $B\left(g\right)$ is called the metamorphic matrix of $A\left(r\right)$. Thus, Equation (13) can be rewritten in the following form:(26)$$\left\{\begin{array}{l} x_{i}\cdot g_{x}-g_{x}\cdot y_{i}=0\\ \left[A\left(x_{i}\prime \right)g_{x}-B\left(y_{i}\prime \right)g_{x}\right]+\left[A\left(x_{i}\right)g_{x}\prime -B\left(y_{i}\right)g_{x}\prime \right]=0 \end{array}\right.$$

Now, minimize the rotation matrix calibration equation in Equation (26), determine the rotation error objective function $f_{1}$, and model the optimization problem using the unit quaternion property as a constraint 1:(27)$$\min f_{1}\left(g_{x}\right)=\sum_{i=1}^{n}||x_{i}\cdot g_{x}-g_{x}\cdot y_{i}|{|}_{F}^{2}\;s.t\;g_{x}^{T}g_{x}=1$$

We determined the relaxation factor to be 2 according to Equation (20) and used the LMI optimization method to find the real part of the dual quaternion $g_{x}$.

For the translational vector calibration equation in Equation (26), organizing it into matrix form yields
(28)$$\left[A\left({x^{\prime }}_{i}\right)-B\left(y_{i}\right)\right]g_{x}+\left[A\left(x_{i}\right)-B\left(y_{i}\right)\right]{g^{\prime }}_{x}=0$$

Then, for i=1,2,⋯,n, describing the relative motion of the subrobot and the camera, there are
(29)$$\left[\begin{array}{c} A\left({x^{\prime }}_{1}\right)-B\left({y^{\prime }}_{1}\right)\\ \vdots \\ A\left({x^{\prime }}_{n}\right)-B\left({y^{\prime }}_{n}\right) \end{array}\right]g_{x}+\left[\begin{array}{c} A\left(x_{1}\right)-B\left(y_{1}\right)\\ \vdots \\ A\left(x_{n}\right)-B\left(y_{n}\right) \end{array}\right]{g^{\prime }}_{x}=0$$

By denoting the 4i×4  matrices in the above equation (in left-to-right order) as *Q*′ and *Q*, Equation (29) can be deformed as follows:(30)$$Q^{\prime }g_{x}+Q{g^{\prime }}_{x}=0$$

By minimizing the equation in Equation (30), we can determine the objective function $f_{2}$, with $g^{T}\cdot g=1,g^{T}\cdot g^{\prime }=0$ constraints, to establish optimization problem model 2:(31)$$\min f\left(g_{x}\prime \right)=\sum_{i=1}^{n}||Q^{\prime }g_{x}+Q{g\prime }_{x}|{|}_{F}^{2}\;s.t\;g_{x}^{T}g_{x}=1\;g_{x}^{T}g_{x}^{\prime }=1\;\;g_{x1}\geq 0$$

When solving an optimization problem, one can consider adding some additional conditions or constraints to limit the solution space and reduce the number of solutions. These additional constraints can be linear or nonlinear, and their introduction can help to exclude some unstable solutions and make the final optimization results more reliable and consistent. Similarly, the convex relaxation optimization method is then used to find the dual part of the dual quaternion (Figure 2).

The steps of the dual quaternion hand–eye calibration algorithm based on convex relaxation optimization proposed in this article are as follows.

Inputs: Robot pose matrix $A_{i}$ and camera pose matrix $B_{i}$ for *i* sets of relative motions.

(1) Considering the solution accuracy and speed requirements, set the iteration accuracy to 0.5 × 10^−20^.

(2) Define polynomial variables *g*_*x*_ and *g*′_*x*_. $g_{x}={\left(g_{x1},g_{x2},g_{x3},g_{x4}\right)}^{T}$, and $g_{x}^{\prime }={\left(g_{x5},g_{x6},g_{x7},g_{x8}\right)}^{T}$.

(3) Establish optimization problem model 1 according to Equation (27). 

(4) Determine the relaxation coefficients, introduce a function $L_{y}$ to linearize the optimization problem, and model the LMI optimization problem as follows:(32)$$\min L_{y}\left[W\left(x\right)\right]\;s.t\;M_{t}\left(y\right)\geq 0\;i=1,\cdots ,k\;M_{t-d}\left(y\right)\geq 0\;i=1,\cdots ,k$$

(5) Solve $msdp\left\{\min L_{y}\left[W\left(x\right)\right],M_{t}\left(y\right),M_{t-d}\left(y\right)\right\}$ to obtain the real part of the dual quaternion $g_{x}$.

(6) Create the optimization problem model 2 according to Equation (31).

(7) Substitute $g_{x}$, and repeat steps 4 and 5 to obtain the dual part of the dual quaternion $g_{x}^{\prime }$.

Output: optimal solution to the hand–eye transformation matrix *X* obtained after global optimization.

## 5. Experiment and Result Analysis

### 5.1. Experimental Environment Construction

In order to further validate the accuracy and robustness of the algorithm proposed in this article, experiments were carried out by installing the Ubuntu operating system on a computer with an Intel(R) Core(TM) i5-10500 CPU opperating @ 3.10 GHz and with 12 GB of RAM, using the ROS operating system version 18.04 Melodic, and performing eye-in-hand calibration experiments. In this experiment, the eye-in-hand method is adopted; namely, the camera is installed at the end of the robotic arm. The robotic arm used in this experiment is an Elfin series Elfin-10 manufactured by Dazu Corporation. It weighs 43 Kg and has an effective workload capacity of 10 Kg, a working range of 1000 mm, a maximum tool speed of 2 m/s, and a repetitive positioning accuracy of ± 0.03 mm. The depth camera used is a RealSense D415i manusfactured by Intel Corporation, with a resolution of 1280 × 720 pixels. A tessellated grid plane target was used for the calibration of the camera parameters, with tessellated grid parameters consiting of dimensions of 8 × 6 and a side length of 24 mm. ArUco code was used for the calibration of the camera parameters, with an ID of 582 and size a of 50 mm. Before the experiment, the camera needed to be first fixed on the manipulator end-effector flange, and the corresponding camera SDK software driver package needed to be installed. At the same time, the stroke and tension of the gripper were tested to ensure the safety of the instrument by adjusting the stroke and tension for different tasks. The specific task of positioning for switching operation studied in this article is illustrated in Figure 3: (a) indicates the positioning of the power instrumentation knob switches on the electric control cabinet; (b) indicates the positioning of the secondary switches in the electric control cabinet; and (c) indicates the positioning of the locking holes on the electric control cabinet.

### 5.2. Data Acquisition

Camera calibration serves as the foundation for position measurement based on visual images, with good camera calibration being essential for improving measurement accuracy. In this experiment, Zhang Zhengyou’s camera calibration method was adopted, in which a checkerboard grid is used as a calibration plate to obtain internal camera reference points. Zhang Zhengyou’s calibration method integrates the advantages of traditional camera calibration methods and camera self-calibration methods. It overcomes the drawbacks of traditional camera calibration methods, which often require high-precision three-dimensional calibrators, and addresses the issues of low precision and poor robustness associated with camera self-calibration methods. With this method, only images of calibration plates at different positions need to be collected, and the pixel coordinates of corner points in the images are extracted. The initial values of the camera’s internal parameters are then calculated using the single-stress matrix, and distortion coefficients are estimated using the nonlinear least-squares method. This approach is not only simple, flexible, and convenient but also offers high calibration accuracy. Currently, it is widely used in solving in-camera parameters. 

Visualization results were obtained, and the camera calibration method calculated the parameters of the checkerboard grid calibration plate at each position, determining the relative position relationship between the camera and the calibration plate at each position, as shown in Figure 4b. The average reprojection error of the tessellated target image is presented in Figure 4a, with each image’s average reprojection error being 0.43 pixels. The calibration results exhibit high accuracy and meet the requirements of the subsequent position measurement experiments. Finally, the internal reference matrix of the camera was calculated as follows:(33)$$A=\left[\begin{array}{ccc} 881.21 & 0 & 215.64\\ 0 & 884.22 & 274.45\\ 0 & 0 & 1 \end{array}\right]\;distortion\;coefficients=\left[\begin{array}{ccccc} -0.02078 & 0.01459 & 0.00450 & -0.01552 & 0 \end{array}\right]$$
where *A* denotes the camera’s internal reference matrix in the derived hand–eye relationship; the distortion coefficients denote the camera’s distortion model coefficients; −0.02078, 0.01459, and 0.00450 are the radial distortion parameters; and −0.01552 and 0 are the tangential distortion parameters.

Via the Ubuntu operating system, Rviz was used to open the robot model, displaying the necessary coordinate changes in the robot’s base coordinate system, camera coordinate system, and end-gripper coordinate system. By using the Rqt command to open the image visualization window, specific information of the ArUco code was identified, facilitating the collection of positional changes in the robotic arm from different angles. The steps of hand-eye calibration are as follows: (1)Manually adjust the robotic arm so that the ArUco code moves to the center of the camera’s field of view. Click ‘check starting pose’; if the check is successful, the interface will display ‘0/17’, indicating that the procedure is ready and can be started.(2)Click ‘Next Pose’, ‘Plan’, and ‘Execute’ in turn. The robot arm will move to a new position. If the ArUco code is within the camera’s field of view and can be detected successfully, proceed to the next step.(3)Click ‘Take Sample’ in Interface 2. If valid information appears in the Samples dialog box, this indicates that the first point calibration is successful.(4)Repeat steps 2 and 3 until all 17 points are calibrated.

By following the above steps, 17 sets of positional data on the robotic arm were obtained. Five sets of data with large errors were eliminated, and finally, 12 sets of data were selected, as shown in Table 1.

In Table 1, Rotation represents the posture of the robot, expressed in quaternions, while Translation represents the position of the robot. The 12 sets of data obtained were subjected to hand–eye calibration using Tsai’s method, Horaud’s method, the dual quaternion method, and the DQCR method (proposed in this article) to perform an error analysis of the experimental results.

### 5.3. Experimental Results and Analysis

Using the collected pose data as an input, the DQCR hand–eye calibration optimization algorithm proposed in this paper was used to solve the hand–eye relationship matrix. The most recently obtained global optimal hand–eye relationship matrix is
(34)$$X=\left[\begin{array}{cccc} 0.02473 & 0.95395 & -0.29817 & -0.17564\\ 0.97552 & -0.08838 & -0.20134 & 0.52855\\ -0.21849 & -0.28662 & -0.93279 & 0.94272\\ 0 & 0 & 0 & 1 \end{array}\right]$$

The transformation matrix of the camera coordinate system with respect to the coordinate system of the end-gripper jaw was obtained using the aforementioned hand–eye transformation matrix. Then, the position of the specific task performed was recognized through end camera recognition, and the position information was converted into the position under the arm’s coordinate system. For the specific tasks mentioned in this article, such as positioning the instrument knob, secondary switch, and locking hole, 50 repetitive positioning experiments were carried out, and the following error accuracy curve was plotted.

As illustrated in Figure 5, after using the Tsai calibration method to localize the aforementioned tasks, the localization error interval was observed between 4 and 4.8 mm, ultimately converging to about 4.4 mm. When employing the Horaud calibration method, the localization error interval ranged between 3.8 and 4.5 mm, finally converging to about 4.0 mm. When employing the Daniilidis calibration method for the tasks, the localization error interval fell between 3.3 and 3.8 mm, ultimately stabilizing at about 3.4 mm. Utilizing the DQCR calibration method yielded a localization error interval between 2.6 and 3.3 mm, eventually converging to about 2.9 mm.

For the identification of the power instrument knob and secondary switch, typically involving the positioning of the center, the space reserved for the end-effector clamping jaws is relatively large, and errors within the range of 5 mm to 1 cm are acceptable. Both traditional hand–eye calibration methods and the optimized DQCR algorithm can effectively complete these tasks. However, in this study, the precision requirements for identifying the lock hole are much higher. With the key inserted into the lock hole, the reserved space measures between 2.5 mm and 3.0 mm. Traditional hand–eye calibration algorithms may not achieve the necessary accuracy for this task. The DQCR optimization algorithm proposed in this article ensures stable positioning accuracy within the 2.9 mm range, meeting the requirements for identifying locking holes in switching operations.

Meanwhile, in order to verify the accuracy of different calibration methods for the localization and recognition of the same object, the following heat map of the four hand–eye calibration methods and the target position was plotted.

As illustrated in Figure 6, the ‘1’ in the center represents the recognition rate of the target location as a percentage. The heat map presented in this article illustrates the correlation between the four hand–eye calibration methods and the recognition rate for the target location. Specifically, the accuracy of Tsai’s hand–eye calibration method in repeatedly recognizing the target object is approximately 79%. Horaud’s hand–eye calibration method achieves an accuracy of around 83% in the repeated recognition of the target object, while Daniilidis’ method achieves an accuracy of around 88%. The DQCR-optimized hand–eye calibration method demonstrates the highest accuracy, at around 91%, for the repeated recognition of target objects. These results indicate that the DQCR-optimized hand–eye calibration algorithm proposed in this article offers a superior recognition rate and positioning accuracy.

Since there is some noise and error in the process of obtaining motion pose matrix A using visual estimation during the hand–eye calibration process, in order to compare the robustness of different methods to the input camera motion pose with perturbation, this experiment adds Gaussian noise with zero mean and a standard deviation of 0.01 d/deg to the rotational part of the camera motion position matrix and Gaussian noise with zero mean and a standard deviation of 0.02 d/mm to the translation part, where d is the noise level (given here as d = 1, 2, …, 10). The relationship between the relative errors of the rotation matrix and translation vectors was simulated under different noise levels, and the line graphs of the rotation error and translation error were plotted as follows. 

Based on the results depicted in Figure 7, it is evident that, under the same noise level, the hand–eye calibration optimization method proposed in this article outperforms other hand–eye calibration methods in terms of both rotational and translational errors while also demonstrating less susceptibility to noise growth. This observation indicates that the estimation results obtained from the algorithm presented in this article exhibit greater stability, further validating the efficacy of solving rotation matrix and translation variables separately, thereby enhancing robustness to noise.

## 6. Conclusions and Perspective

Acquiring more accurate positional information is crucial to allow robots to execute tasks since traditional optimization algorithms rely on the choice of initial values and are prone to converge to local optimal solutions. To address the above problems, this article proposes a hand–eye calibration algorithm based on convex relaxation for global optimization. By employing the mathematical concept of dual quaternions, the optimal pair of values was obtained through calculation; at the same time, the optimality of the solution is guaranteed without requiring initial value estimation. The obtained results show that the global optimization algorithm proposed in this article has higher accuracy and robustness than the nonlinear optimization hand–eye calibration algorithm, not only accomplishing fundamental tasks like turning knobs, flipping switches, and performing other operations but also more intricate tasks such as inserting and extracting keys during switching operations. It holds promise for future applications in high-precision robotic vision systems. Due to the limited experimental conditions, this article only focuses on the current laboratory environment. The subsequent plan is to test the data in different environments and scenarios, check whether the algorithm has sufficient adaptability to allow it to be improved, further enhance the algorithm’s processing speed, and broaden the algorithm’s range of applications.

## Figures and Tables

**Figure 1 sensors-24-03830-f001:**
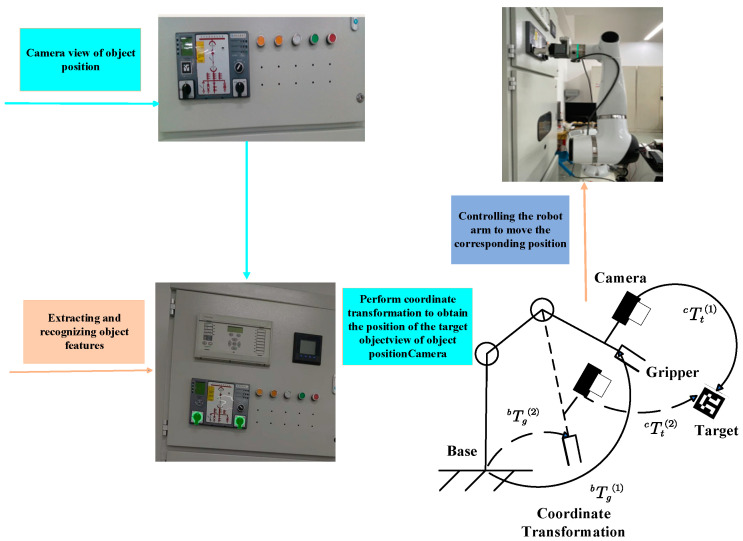
Structure of the switching operation system.

**Figure 2 sensors-24-03830-f002:**
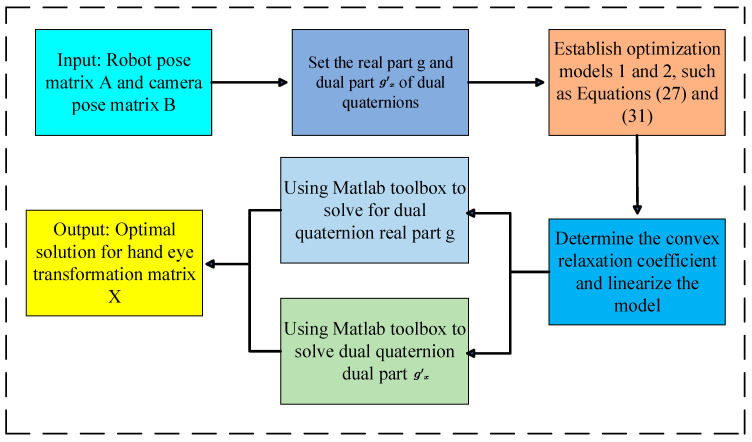
Flow chart for solving the hand–eye relationship matrix.

**Figure 3 sensors-24-03830-f003:**
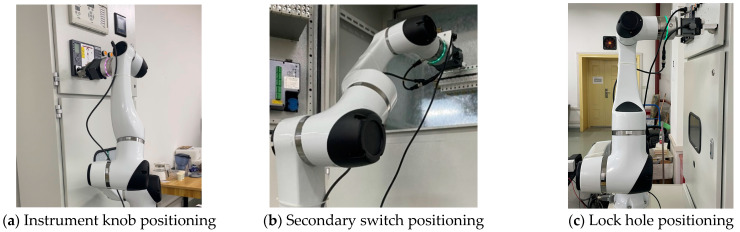
Specific task execution diagram for gate reversal operation.

**Figure 4 sensors-24-03830-f004:**
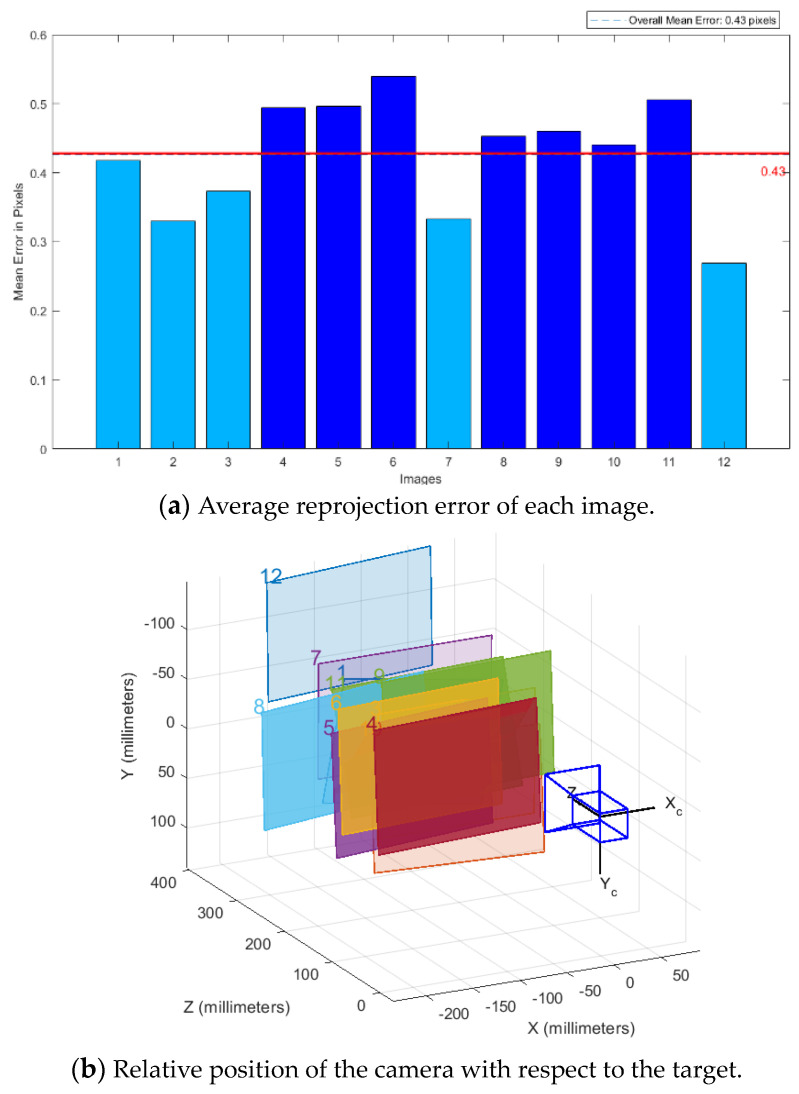
Camera internal reference calibration.

**Figure 5 sensors-24-03830-f005:**
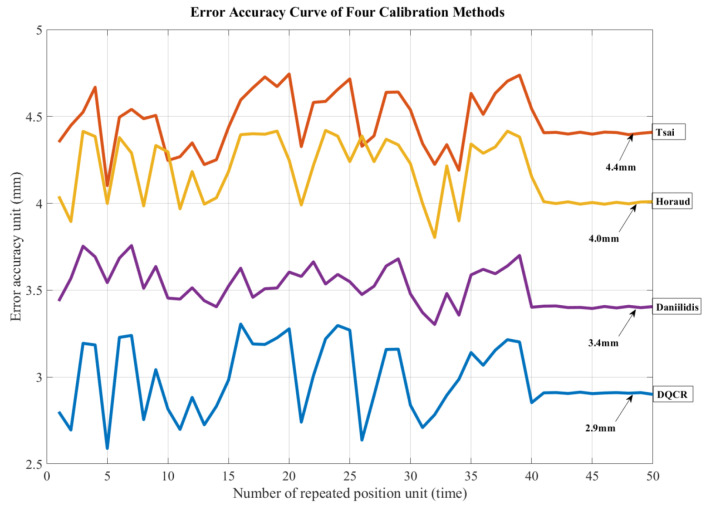
Error accuracy curves of the four calibration methods.

**Figure 6 sensors-24-03830-f006:**
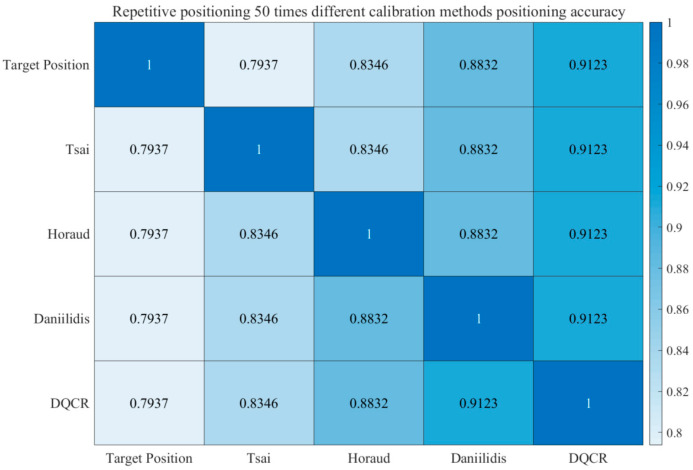
Comparison of repeat localization heat maps for different calibration methods.

**Figure 7 sensors-24-03830-f007:**
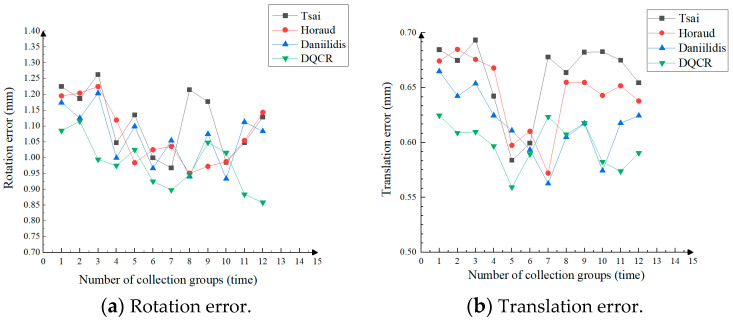
Rotation and translation error plots of adding noise level to hand–eye calibration.

**Table 1 sensors-24-03830-t001:** The 12 sets of positional data on the target object relative to the camera.

Groups		Rotation (x,y,z,w)				Translation (x,y,z)	
1	−0.24	0.4	0.55	0.05	0.04	−0.01	0.65
2	0.05	0.03	0.39	0.74	0.66	−0.14	0.07
3	−0.45	0.42	0.55	0.06	0.06	0.22	0.84
4	−0.04	0.11	0.44	0.7	0.59	−0.11	0.39
5	−0.34	0.42	0.31	−0.17	−0.06	0.01	0.92
6	−0.26	0.35	0.51	0.19	0.27	−0.03	0.81
7	−0.08	0.15	0.45	−0.66	−0.64	0.12	0.39
8	−0.25	0.35	0.53	0.2	0.17	−0.03	0.78
9	−0.33	0.49	0.55	0.07	0.07	−0.01	0.68
10	0.02	0.05	0.41	0.72	0.65	−0.13	0.19
11	0.12	0.08	0.43	0.16	0.58	−0.23	0.14
12	−0.23	0.42	0.39	−0.12	0.14	−0.32	0.44

## Data Availability

The data presented in this study are available on request from the corresponding author.

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
