# Peer review of "A High-Precision Hand–Eye Coordination Localization Method under Convex Relaxation Optimization"

_sensors, 2024, doi:10.3390/s24123830_

Round 1

Reviewer 1 Report

Comments and Suggestions for Authors

The authors wrote an article about A high-precision hand-eye coordination localization method under convex relaxation optimization. The article is well put together and contains all the essentials. I like the use of math. However, I have a few minor comments:

- Citations of sources in the text must be numbered consecutively

- Why are the numbers 1, 2 and 3 on lines 117, 122 and 127?

- Fig. 1 has different fonts. It would be nicer to unify it.

- On line 217, ijk=-1. Is it really true?

- Chinese characters remained not only on line 286, 288. Go through the entire article.

- There is a wrong chapter number on line 398

- Fig. 3 is too small, unreadable data

- Why is the number 4 on line 440?

- Lines 545-550 are too big

- Some citations are missing in the text.

Reviewer 2 Report

Comments and Suggestions for Authors

The research presents a new high-precision hand-eye localization method through the use of a dual quaternions tool and linear matrix inequality convex optimization method, obtaining adequate results compared to conventional methods.

- Figure 1 has an error in the title. Furthermore, there is no agreement with the description or clarity.

- In some expressions a symbols appear that are not described or were included by mistake, for example: equations (17) and (23).

- Reference is not included for Lasserre, line 294.

- The performance of the solvers is not considered in the solution. Each has a particular condition.

- Present the description of the proposed algorithm as pseudocode and flowchart. Also, clarify if the solution is carried out offline or online.

- In Figure 3(a), the average projection error is presented, but in the text, it indicates a reprojection error, clarify.

- Clarify the values of each group presented in Table 1.

- Finally, review in detail the editing and presentation of the document. There are different font sizes, titles of figures, and inadequate distribution of information, among others.

Reviewer 3 Report

Comments and Suggestions for Authors

The manuscript proposes an approach to hand-eye calibration in switch operation tasks within power distribution rooms, leveraging a dual quaternion-based convex relaxation global optimization algorithm. And the manuscript claims significant improvements in accuracy and robustness over existing methods. While very similar algorithms have been investigated in other published works, the literature review is less comprehensive and the references from recent years are required to be considered.

The convex relaxation approach for global optimization is reasonable, and the methodology is clear and logical progression from problem formulation to solution implementation.  While a pseudocode or flowchart of the algorithm is encouraged to enhance the clarity. 

The design of the experiment involves comprehensive comparisons with existing hand-eye calibration methods. The results, presented through various metrics, convincingly demonstrate the performance of the algorithm in terms of rotation and translation error reduction. However, the experiments are primarily conducted in a controlled laboratory setting. And the computational complexity and time efficiency of the proposed algorithm should be discussed.

Some figures and tables are blurry.

Comments on the Quality of English Language

The paper is poorly written in English, even with non-English characters appearing in lines 286, 288 and 313.

Round 2

Reviewer 3 Report

Comments and Suggestions for Authors

I am grateful for the responses, but I do not think the quality of the paper has been sufficiently improved. Overall, the prose is clear and the reviewer had little trouble understanding the authors' meaning.  However, there are many grammatical errors, nonstandard uses of words. The reviewer recommends the authors seek the input of an expert English-language writer to edit the manuscript for grammar.

Comments on the Quality of English Language

The reviewer recommends the authors seek input from an expert writer to edit the manuscript.

Author Response

Thank you for your comments and feedback on this article:
          I have sought professional English writers to check and correct some grammar related issues in the paper you mentioned, as well as to improve the overall content of the article.